# Multi-Dimensional Mathematical Wear Models of Vibration Generated by Rolling Ball Bearings Made of AISI 52100 Bearing Steel

**DOI:** 10.3390/ma13235440

**Published:** 2020-11-29

**Authors:** Paweł Zmarzły

**Affiliations:** Department of Mechanical Engineering and Metrology, Kielce University of Technology, al. Tysiąclecia Państwa Polskiego 7, 25-314 Kielce, Poland; pzmarzly@tu.kielce.pl; Tel.: +48-41-342-44-53

**Keywords:** rolling bearings, AISI 52100, vibration, anderon unit, roundness, waviness, radial clearance, wear model

## Abstract

The paper features the development of multi-dimensional mathematical models used for evaluating the impact of selected factors on the vibration generated by 6304ZZ type rolling ball bearings from three manufacturers in the aspect of the wear process. The bearings were manufactured of AISI 52100 bearing steel. The analyzed factors included the inner and outer raceways’ roundness and waviness deviations, radial clearance and the total curvature ratio. The models were developed for vibration recorded in three frequency ranges: 50–300 Hz, 300–1800 Hz and 1800–10,000 Hz. The paper includes a specification of the principles of operation of innovative measuring systems intended for testing bearing vibration, raceway geometries and radial clearance. Furthermore, it features a specification of particular stages of the multi-dimensional mathematical models’ development and verification. Testing with the purpose of statistical evaluation of the analyzed factors is also presented. The test results and mathematical models indicate that the inner raceway’s waviness deviation had a dominant impact on the vibration examined in all frequencies. The roundness and waviness deviation of bearing raceways made of AISI 52100 steel propagates the bearing wear process.

## 1. Introduction

Due to their properties and standardized nature, rolling bearings are used in almost all technical fields (automotive industry, machine industry, aviation industry, household goods industry, etc.) The structure of rolling ball bearings has not changed for many decades. The rolling element in rolling ball bearings is located between two rings. However, despite the standardized structure, rolling bearing manufacturers try to implement solutions to provide their products with a long service life [1], low noise [2], high stiffness [3] and simultaneous minimization of manufacturing costs, which surely affects the products’ competitiveness.

Rolling bearings are the subject of many scientific and industrial research. The research mainly concerns the search for solutions that enable improving the rolling bearings’ operating parameters. Furthermore, there are many papers on diagnostic methods that evaluate the condition of new and operating bearings [4,5]. It is necessary to mention that rolling bearings in many cases are a critical element in a mechanism. In case of an unexpected bearing or bearing assembly failure, the entire machine is shutdown, which entails high costs.

One of the most important indicators determining the wear of rolling bearings is the vibration generated by them [6]. During its operation, a bearing undergoes wear, which can lead to a gradual increase in generated vibration. The vibration diagnostics, which is a non-destructive procedure, allows for evaluating the tested bearing’s wear. Based on the vibration analysis, it is possible to introduce preventive treatment aimed at planned bearing replacement. This allows for the elimination of unplanned machine shutdown and minimizes repair costs. An analysis of the vibration generated by new rolling bearings also enables the detection of damage or factory defects. Moreover, excessive vibration generated by the bearing is an adverse phenomenon, because aside from a drastic reduction in the bearing’s service life, it has an adverse effect on the entire bearing assembly.

There are many factors that affect the vibration generated by rolling bearings. Despite the fact that there are many papers related to the analysis of the impact of specific factors on the vibration of rolling bearings, there is a lack in statistical evaluation of the significance of the factors’ impact.

Paper [7] presents the mathematical model for the rolling ball bearings’ friction moment with consideration of the bearings’ imperfections, such as waviness. The model was verified experimentally. The mathematical models can also be used to predict the rolling bearings’ service life. In paper [8], Wang et al. presented a mathematical model (matrix) in which the first dimension is the bearing’s service life given in hours and the other—performance. The mathematical model allowed for specifying the bearing’s condition after a specific operation cycle. The model was verified experimentally. On the other hand, paper [9] presents the dynamic model of rolling bearing vibration generated due to the interaction between the rolling elements and the defect area. The paper proves that the raceway’s defect causes a double vibration impulse. This can be an indicator of raceway damage when analyzing the vibration of rolling bearings.

Vibration occurring during the operation of rolling bearings is a natural phenomenon and results directly from the rolling bearings’ structure. Natural vibration is generated during the turning of rolling elements. If it does not exceed the acceptable values, then such bearings are approved.

The testing of rolling bearing vibration can be classified into three groups. The first group concerns the evaluation of the vibration of new rolling bearings on testing rigs. The evaluation is aimed at detecting factory defects of new bearings and is commonly conducted by companies that manufacture rolling bearings by using industrial Anderon meters. This approach allows for streamlining the rolling bearings’ manufacturing and introducing a modification to their structure to meet the costumer’s requirements. The second testing group concerns the vibration analysis of rolling bearings operating in real application conditions, e.g., in electric engines [10], etc. A certain inconvenience of this approach is the risk of transferring the vibration from the entire mechanism in which the bearing was used. In such a case, it is possible to obtain “false” vibration signals leading to erroneous conclusions. The third testing group concerns intentional induction of defects or damage in rolling bearing elements to determine their impact on the generated vibration. The testing is mainly aimed at developing new diagnostic methods for rolling bearings with the purpose of detecting specific defects.

The surface texture quality is one of the most important factors affecting the tribological performance and wear of mechanisms [11,12]. Paper [13] presents the testing of the impact of 2D and 3D roughness parameters of rolling ball bearings’ inner ring raceway on the generated vibration. The raceway surface topography testing was conducted with the use of a focus microscope. The authors demonstrated a high correlation between the roughness parameters Ra and Sa, and the generated vibration. Similar testing was conducted by Adamczak et al. in paper [14]. The evaluation of the impact of bearing raceway surface roughness on the generated vibration was conducted using 4 different roughness parameters (Ra (Sa), Rt (St), Rku (Sku) and Rsk (Ssk)). In the paper, the authors also obtained a high correlation between the inner raceway’s roughness parameters Ra and Sa, and the generated vibration. The testing allowed for the conclusion that in the case of stable rolling bearing manufacturing, it is sufficient to analyze the 2D raceway surface roughness parameters. On the other hand, in the case of distorted manufacturing or production of new bearing types, it is recommended to conduct a detailed analysis of the raceway’s topography.

It is necessary to add that the roughness profile was usually measured in the described scientific papers along a single raceway cross-section. On the other hand, due to the balls’ rolling motion on the raceway’s entire surface, it seems that a detailed analysis of the raceway’s surface is more reasonable, i.e., by evaluation the waviness and roundness deviations.

One of the dominant factors affecting the generated vibration is the raceway’s waviness. Paper [15] features a simulation of the occurrence of waviness on rolling ball bearing rings and evaluation of its impact on the bearing’s behavior. It was demonstrated that the raceway’s waviness causes a non-linear vibration response. The paper lacked an experimental verification of the simulation tests. Similar testing was conducted by Wang et al. [16]. In paper [17], the authors applied the adopting signal coherence theory to evaluate the impact of the raceways surface waviness on the generated vibration. They demonstrated that the outer ring raceway’s waviness is more strongly correlated with the vibration analyzed in a low range of frequencies.

It is necessary to add that most papers only featured an analysis of the waviness’ impact, whereas a different type of deviations, especially long-wave deviations such as roundness is omitted. It is a certain simplification, because the ball does not only roll on the raceway’s roughness or waviness but also on “all deviations” [18]. In such a case, it is necessary to apply filtration methods and measurement signal analyses [19,20]. Industrial practice usually features testing of the outer ring’s roundness, because its excess value contributes to issues with the bearings’ mounting. Viitala et al. [21] analyzed the impact of the bearing’s inner ring roundness profile on the sub-critical vibrations in a flexible rotor. They demonstrated a significant correlation between the tested bearing’s inner ring roundness and the generated vibration. Minimization of the inner ring’s roundness deviation contributed to the reduction in the tested mechanism’s vibration. Interesting research was conducted in Japan [22]. The method of mounting the bearing also affects the vibration generated by rolling bearings. One of the basic structural parameters of rolling ball bearings is the radial clearance. Paper [23] presents an online system intended for evaluating the bearing clearance based on the vibration signal. In paper [24], Zmarzły presented testing that demonstrated increased vibration as result of increasing the radial clearance.

Another parameter that describes the inner geometry of rolling ball bearings is the curvature ratio. Gloeckner [25] tested the impact of the curvature ratio on the temperature and energy losses in bearings used in jet engines. However, the number of research papers related to the evaluation of the ratio’s impact on rolling bearing vibration is limited. Only the author of [26] has conducted an analysis of the inner and outer raceway’s curvature ratio in relation to the vibration generated by the 6304 type bearings.

A novelty of the article is development multi-dimensional mathematical models used to quantitative analysis of the impact of selected factors on vibrations generated by rolling bearings made of AISI 52100 steel. A literature analysis demonstrated that the Authors are focusing mainly on testing the impact of single factors on the vibration of rolling bearings recorded in a full range of frequencies. It is a certain limitation, because the impact of some factors can be clearer in a narrowed range of frequencies. Due to the above, it was proposed in industrial practice to analyze vibration generated by bearings in 3 frequency ranges, i.e., low LB 50–300 Hz, medium 300–1000 Hz and high 1000–18,000 Hz. However, there is a shortage of papers that specify quantitatively the manner in which a given factor affects vibration recorded in a specific frequency range and whether this impact is relevant. It should be noted that it is an issue to clearly specify the factors that are dominant in their impact on the vibration generated by rolling bearings. Therefore, it is recommended to conduct a quantitative evaluation of the factors’ impact. Such evaluation can be conducted by using mathematical models developed on the basis of multiple regression. Furthermore, in most of the analyzed scientific papers, the testing is limited to solely computer simulation tests or experimental tests on a small group of rolling bearings from a single manufacturer. Despite the fact that the manufacturing of rolling bearings is a common and widely known process, bearing manufacturers apply various company secret processes to obtain products with satisfying quality. This is the reason for the large price and quality disproportion between bearings of the same type but from different manufacturers.

The main objective of the paper is a complex analysis of the impact of a selected factor on vibration level of 3 groups of rolling bearings offered by three different manufacturers and the development of mathematical models to quantitatively evaluate the significance of this impact. The models can be used to estimate the vibration generated by specific bearings based on the measured performance and structural parameters.

## 2. Materials and Methods

### 2.1. Proposed Method

The literature analysis presented in chapter 1 demonstrated that there are many factors that affect the generated vibration to a greater or smaller degree. However, most papers lack detailed statistical testing aimed at conducting a quantitative evaluation of the impact of selected factors on vibration that is recorded in specific frequency bands. The literature analysis and own testing conducted by the author of [24,26,27] serves to distinguish 5 factors that have a substantial impact on the generated vibration, which include inner and outer raceway roundness and waviness, total curvature ratio and radial clearance. These are the parameters that sufficiently describe the geometry and quality of the surface layer of key bearing elements, specifically of the outer and inner raceway. Three mathematical models were developed for each vibration frequency band, i.e., low (LB), medium (MB) and high (HB). Due to the fact that the analysis covered 5 different variables dependent on the generated vibration, the experiment planning methods should make the minimum number of samples amount to n = 2^5^ = 64. It is necessary to add that a higher number of samples allows for a more reliable analysis. Due to the above, a total of 90 rolling bearings were tested (3 samples of 30 bearings from different manufacturers). All bearings were manufactured of AISI 52100 bearing steel.

One of the methods to conduct a quantitative evaluation of the impact of many factors on the tested parameter is statistical testing consisting of the development of multi-dimensional regression models. For this purpose, a linear multiple regression, which allows for the simultaneous analysis of many independent variables, was conducted. The general linear multiple regression model can be described using the Equation (1).
(1)$$y=\beta_{0}+\beta_{1}X_{1}+\beta_{2}X_{2}+\ldots +\beta_{k}X_{k}+\epsilon$$
where
y—dependent variable (explained variable);x_1_, x_2_, …, x_k_—independent variables (predictors);$\beta_{0}$—regression constant,$\beta_{1},{\;\beta }_{2},\ldots ,\beta_{k}$—regression function’s structural parameters;ϵ—random component (model error).

When considering Equation (1), it is possible to notice that the dependent variable is affected by the linear combination of the (tested) independent variables, i.e., x_1_, x_2_, …, x_k_, and the random component that derives from the impact of factors not analyzed in the model or calculation errors. Hence the name of the model error. At a sufficient number of results of measurement n, Equation (1) can be written in the form of a matrix Equation (2):(2)Y=Xβ+ϵ
then
$$Y=\left(\begin{array}{c} y_{1}\\ y_{2}\\ \vdots \\ y_{k} \end{array}\right),\;X=\left(\begin{array}{ccccc} 1 & x_{11} & x_{21} & \cdots & x_{k1}\\ 1 & x_{12} & x_{22} & \cdots & x_{k2}\\ \vdots & \vdots & \vdots & \ddots & \vdots \\ 1 & x_{1k} & x_{2k} & \cdots & x_{\mathrm{kn}} \end{array}\right),\;\beta =\left(\begin{array}{c} \beta_{0}\\ \beta_{1}\\ \beta_{2}\\ \vdots \\ \beta_{k} \end{array}\right),\;\epsilon =\left(\begin{array}{c} \epsilon_{1}\\ \epsilon_{2}\\ \vdots \\ \epsilon_{n} \end{array}\right).$$

The solution to the above equation is the vector of the evaluation of the regression function’s structural parameters Equation (3).
(3)$$b=\left(\begin{array}{c} b_{0}\\ b_{1}\\ b_{2}\\ \vdots \\ b_{k} \end{array}\right)$$

Then, the estimated regression function will take the following form:(4)$$y=b_{0}+b_{1}X_{1}+b_{2}X_{2}+\ldots +b_{k}X_{k}+\epsilon$$

The estimation of the regression coefficients is usually performed by using the least squares method in order to obtain parameters for which the sum of the regression’s squared residuals will be smallest. In practice, it is necessary to minimize Equation (5).
(5)$$\sum_{i=1}^{n}{\left(y_{i}-{\hat{y}}_{i}\right)}^{2}$$

Each regression coefficient estimation is burdened with an estimation error that can be designated from Equation (6):(6)$${\mathrm{SE}}_{b}=\sqrt{\frac{1}{n-\left(k+1\right)}e^{T}e{\left(X^{T}X\right)}^{-1}}$$
where
n—number of observations;k—number of estimated model parameters;e—model residuals’ vector.

On the other hand, the estimate’s standard error, which is the entire model’s match measure, can be specified by Equation (7).
(7)$${\mathrm{SE}}_{e}=\sqrt{\frac{\sum_{i=1}^{n}e_{i}^{2}}{n-\left(k+1\right)}}$$

The paper features an analysis of the impact of six factors affecting the generated vibration. Therefore, the model includes six predictors. Based on Equation (4), the general regression model presenting the impact of selected factors on the generated vibration will take the following form:(8)$$y=b_{0}+b_{1}{\mathrm{RONt}}_{\mathrm{PW}:\left(2-15\right)}+b_{2}{\mathrm{RONt}}_{\mathrm{PZ}:\left(2-15\right)}+b_{3}{\mathrm{RONt}}_{\mathrm{PW}:\left(16-50\right)}+b_{4}{\mathrm{RONt}}_{\mathrm{PZ}:\left(16-50\right)}+{\;b}_{5}\Delta R+\;+{\;b}_{6}f_{t}+{\mathrm{SE}}_{e}$$
where
b_0_—free expression;b_1_, b_2_, b_3_, b_4_, b_5_, b_6_—regression coefficient;${\mathrm{RONt}}_{\mathrm{PW}:\left(2-15\right)}$—inner ring raceway’s roundness deviation (2–15 upr);${\mathrm{RONt}}_{\mathrm{PZ}:\left(2-15\right)}$—outer ring raceway’s roundness deviation (2–15 upr);${\mathrm{RONt}}_{\mathrm{PW}:\left(16-50\right)}$—inner ring raceway’s waviness deviation (16–50 upr);${\mathrm{RONt}}_{\mathrm{PZ}:\left(16-50\right)}$—outer ring raceway’s waviness deviation (16–50 upr);ΔR—radial clearance;f_t_—total curvature ratio.

One of the factors included in model described by Equation (8) were the RONt roundness deviations interpreted as the sum of the highest positive and the absolute highest negative value of the local roundness deviation determined for the LSC reference circle. However, the parameters were analyzed in two ranges of sinusoidal undulations per one element revolution (UPR). The first range covers 2–15 UPR and the parameter was designated as RONt_2–15_. On the other hand, the second parameter covers 16–50 UPR and was designated as RONt_16–50_. The filtration was performed using the Gaussian filter with the limit undulation length of λ_c_ = 0.8. It is necessary to add that despite the fact that RONt is determined as the roundness deviation, the use of the high pass filter f_c_ = 15 UPR allows for filtering the roundness components and saving the waviness components. Due to the above, the RONt_16–50_ parameter started to be referred to as the waviness deviation of cylindrical surfaces. It should be mentioned that roundness and waviness deviation have been examined for inner and outer bearing rings. Therefore, $RONt_{PW}$ described parameters calculated for inner raceways, but $RONt_{PZ}$ indicated parameters calculated for outer raceways. Figure 1 presents a schematic presentation of the difference between the outline of the roundness and waviness of rolling bearing raceways with the existence of ovality on the rings.

Another factor included in the regression model was internal radial clearance ΔR that shows the total radial movement of the outer ring with respect to the inner ring in a plane perpendicular to the bearing axis. The radial clearance affects vibration generated by bearing. The last factor presented in model Equation (8) was total curvature radio f_t_ that described contact area of balls and bearing raceways.

It is possible to conduct statistical testing in order to evaluate whether a given predictor in the analyzed mathematical model (see Equation (8)) is statistically relevant and substantially affects the dependent variable. One of the tests that can be applied is the so-called Student’s *t*-test. Then, it is necessary to calculate the statistics value from Equation (9).
(9)$$t=\frac{b_{i}}{{\mathrm{SE}}_{b}}$$

It is necessary to deem the analyzed test statistic to have the student t-distribution. Then, it is necessary to read t_kr_ = t_α,n−k−1_ from the tables. If we assume the significance of α = 0.05 for particular predictors presented in Equation (8), the critical value will take the following form:t_kr_ = 1.989(10)

If $\left|t\right|>t_{\mathrm{kr}}$, then it is possible to state that the given variable is statistically significant and it must be taken into account in the model. Otherwise, it is necessary to deem the variable as not substantially affecting the dependent variable and it must be removed from the model.

The statistical significance of the entire model can be evaluated by using the so-called Fisher’s F-test. The statistic F is calculated from Equation (11):(11)$$F=\frac{\frac{H_{\mathrm{ss}}}{k}}{\frac{E_{\mathrm{ss}}}{n-\left(k+1\right)}}$$
where
H_ss_—sum of squares explained by the model;E_ss_—residual sum of squares;n—number of observations;k—number of estimated model parameters.

The critical value must be read from the tables for F_kr_ = F_α,k,n−k−1_. Similarly, as for the student *t*-test, a significance of α = 0.05 was selected for the F-test. Then, the critical value for the model described by the Equation (8) is as follows:F_kr_ = 2.33(12)

If F > F_kr_, then the entire model can be deemed as statistically significant.

Another important coefficient describing the quality of the developed model is the multiple determination coefficient R^2^. It expresses the percentage variation of the dependent variable expressed by the model. In practice, it is the model’s matching degree. For example, if we obtain R^2^ =0.6, then 60% of the original variation y was expressed by the regression equation. The determination coefficient R^2^ can be designated using Equation (13).
(13)$$R^{2}=\frac{T_{\mathrm{ss}}}{H_{\mathrm{ss}}}\;\;$$
where
T_ss_—total sum of squares;H_ss_—sum of squares explained by the model.

The assumptions and dependencies presented above were used for the statistical analysis of the obtained results to develop the mathematical regression models describing the impact of selected factors on the generated vibration.

### 2.2. Experimental Testing

The testing conducted for the purpose of the paper included multiple stages. Prior to testing commissioning, rolling bearings were adequately grouped and marked. The testing was conducted using one of the more popular rolling ball bearings available in the market, specifically the 6304ZZ type bearings. These are bearings sealed with plates, thereby eliminating the risk of contaminating the lubricating agent. Furthermore, during the bearings’ operation, the protective plates provide protection against uncontrolled grease spillage. Table 1 presents the basic operating and structural parameters of the tested bearings.

Due to the fact that the test procedures, the effect of which is the mathematical models’ development, consisted of several stages, they were presented in a block diagram presented in Figure 2.

One of the main and first stages of testing was the vibration analysis of rolling bearings, the parameters of which are presented in Table 1. The Anderonmeter measurement system was used for this purpose. Its structure and operating principle is similar to anderon meters commonly used in industrial conditions for the purpose of quickly inspecting rolling bearing vibration. However, most industrial anderon meters operates following the principle of “good” or “not good”. This means that after exceeding the acceptable vibration characteristics, a bearing is discarded or transferred for further analysis. Due to the above, the Anderonmeter system software was developed to enable a detailed vibration analysis of the tested bearings [28,29]. Figure 3 presents a picture of the Anderonmeter instrument’s test head along with the tested bearing.

The Anderonmeter measurement system is equipped with a set of replaceable retaining rollers (1), the dimensions of which are adapted to the bearing’s inner diameter. The roller (1) is embedded in a spindle (2) with hydrodynamic plain bearing lubricated with oil mist from exterior pomp. The use of the spindle’s hydrodynamic plain bearing allows for eliminating the impact of own vibration on the measurement result. The tested bearing (3) is embedded on the roller (1). Stable embedding of the bearing on the roller and elimination of axial clearance are ensured by the pneumatic clamp (4). During the test, the roller (1) sets in revolving motion the bearing’s inner ring via the spindle (2). The inner ring, which is compliant with the standards concerning the rolling bearings’ vibration measurements, is set into revolving motion of 1800 rpm. This is an acceptable value for all of the tested bearing groups (see Table 1). As result of the bearing’s operation, the generated vibration is transferred to the outer ring, which is in contact with the vibration sensor (5) working in velocity function. This type of measuring sensor allows for accurate measurement of low frequencies (below 50 Hz) and the possibility of working with a very low signal amplification, which significantly simplifies the construction of the amplifier. Moreover, the sensor has a low input impedance, which has a positive effect on the suppression of external noise.

One of the factors that substantially affect the rolling bearings’ vibration measurement result is the selection of the measurement point on the outer ring. Due to the above, three vibration measurements every 120° were conducted for each bearing side. A total of six vibration measurements were conducted for each bearing and then the average values were designated.

The piezoelectric sensor’s signal is suitably amplified by the amplifier, which features 7 automatic amplification degrees, i.e., from 250 to 25,000 times. This amplification enables conducting measurements at high resolutions. Furthermore, the amplifier is equipped with its own power source (gel battery), which enables eliminating interference from the electric grid. After adequate amplification of the vibration signal, the software filters it to three frequency bands: 50–300 Hz (low), 300–1800 Hz (medium) and 1800 Hz–10,000 Hz (high). Aside from the standard analysis of the rolling bearings’ vibration expressed with “Anderon” units, the software ensures a Fast Fourier analysis of the vibration signal and envelope analysis. Furthermore, the system is equipped with speakers that enable an acoustic evaluation of the bearings’ operation.

Following the vibration measurement, the next step was to directly measure the radial clearance ΔR by using the station specified in publication [24]. After the radial clearance measurement, the bearings were disassembled to obtain access to the balls and raceways. The protective plates were removed for this purpose. The rivets combining the cages were removed using a pneumatic press. Following the separation of the rings and balls to remove the residue and greases, they were cleaned in an ultrasonic cleaner by using a detergent.

The next step of the test procedure was to designate the total curvature ratio. The bearing raceway’s curvature ratio is determined as the ratio of the raceway radius and the ball diameter. It describes the fitting degree of the rolling elements to the raceway surface in a quantitative manner. There is the curvature ratio of the outer raceway (f_o_) and of the inner raceway (f_i_). On the other hand, the total curvature ratio (f_t_) is the sum of the outer and inner raceway’s curvature ratio. It can be expressed by using the Equation (14).
(14)$$f_{t}=f_{o}+f_{i}-1=\frac{r_{o}}{D_{w}}+\frac{r_{i}}{D_{w}}-1$$
where
r_o_—outer raceway radii;r_i_—inner raceway radii;D_w_—ball diameter.

When analyzing the Equation (14), it is possible to notice that the designation of the total curvature ratio requires measuring the ball diameters and raceway radii. The ball diameters were measured by using a horizontal metroscope. Each tested bearing was equipped with seven balls, due to which the curvature ratio calculation for each bearing utilized the average ball measurement value. On the other hand, the raceway radii were measured by using the STPP measurement system, which was specifically designed at the Kielce University of Technology to analyze the geometry of the rolling bearing ring raceways [26]. Figure 4 presents the instrument’s picture along with the measured outer ring.

A definite advantage of the STPP measurement system is the ability to measure the raceway geometry with the measuring tip that moves along an arch with a specific curvature. Due to the above, the measurement is conducted at the polar coordinates and not the Cartesian coordinates as in the case of most contour measuring instruments. Because of this, the measuring tip is always positioned perpendicularly to the measured surface. This allows for improving the measurement accuracy [26].

The last step of the experimental testing was the evaluation of the raceway’s roundness and waviness deviation by using the Talyrond 365 measuring instrument. It is a precise system based on the principle of radius changes with a revolving table.

## 3. Results and Discussions

The experimental testing results were the basis for conducting statistical testing aimed at designating multi-dimensional regression models. Three regression models were developed for each vibration frequency band (low—LB, medium—MB and high—HB). The statistical calculations for a sample of 90 pieces of rolling bearings were conducted using the “multiple regression” package of the statistics software. The dependent variables (predictors) were as follows: inner ring’s roundness deviation (RONt_PW(2–15)_)_,_ outer ring’s roundness deviation (RONt_PW:(16–50)_), outer ring’s roundness deviation (RONt_PZ:(2–15)_), outer ring’s waviness deviation (RONt_PZ:(16–50)_), radial clearance (ΔR) and raceways’ total curvature ratio (f_t_).

Fisher’s variance test was conducted in order to evaluate the variance of the entire mathematical model. On the other hand, the student t-significance test was conducted for particular input variables. Table 2 presents the analysis results for the low frequency band, i.e., 50–300 Hz, where b indicated regression coefficient, SE_b_ indicated estimation error for regression coefficient, t(83) present “t” statistic, *p*—probability value and β—regression function’s structural parameter.

For the entire mathematical model designated for the low vibration frequency band (LB) the determination coefficient of R^2^ = 0.46 was obtained, whereas the statistics value is F(6.83) = 11.3777 > F_kr_, which means that the model is statistically significant and was developed correctly. This is also confirmed by the value of coefficient p which amounted to *p* < 0.05. The standard estimation error for the entire model amounted to SE_e_ = 1.197. The relatively small value of the determination coefficient R^2^ indicates that there may be other factors (not analyzed in the tests) affecting the vibration values recorded in this frequency band. It should be noted that that the value of vibrations generated in the low frequency range is influenced by difficult-to-measure factors, such as cage unbalance, contamination of the lubricant, etc.

Table 2 presents the statistically significant predictors marked in red. It is possible to see that in this case there are significant input factors that affect the vibration generated in the low frequency range. These are the inner ring’s waviness deviation, roundness deviation and the outer ring’s waviness deviation. The highest value of standardized regression coefficient β was obtained for the inner ring’s roundness deviation. Due to the above, it is possible to assume that the quality of the inner raceway’s surface layer has a substantial impact on the vibration generated in the low frequency band. A similarly high β value was obtained for the outer ring raceway’s waviness. Analyzing calculation result presented in Table 2, it can be concluded that only three input factors among six have substantial impact at vibration generated in low frequency band. Therefore, this factors will be presented on the charts in detail. The impact of the inner raceway’s roundness and waviness deviations on the vibration generated in the low frequency range is presented in a single chart (Figure 5), because they concern the same rings. On the other hand, the impact of the outer ring raceway’s waviness is presented on a separate chart. The vibration level is described by Anderon Unit (And).

When analyzing the chart presented in Figure 5, it can be clearly stated that the increase in the inner ring race’s roundness and waviness deviation contributes to increased vibration tested in the frequency range of 50–300 Hz. Wherein, the waviness was higher than the roundness deviations measured for the inner ring’s race (see Figure 5). A similar dependency can be read for the outer ring. Here, the increase in the waviness deviation RONt_PZ:(16–50)_ causes a moderate increase in the vibration recorded in the low frequency range (see Figure 6). As shown in Figure 1, waviness deviation is periodic or non-periodic irregularities appearing on bearing raceway profile. The largest waviness deviation indicated highest wave’s amplitude on the raceway profile. Therefore, during working, bearing balls generate higher vibration value, which is also recorded in the low frequency range.

When analyzing the mathematical model designated for the medium vibration frequency band (MB), the obtained determination coefficient amounted to R^2^ = 0.44. On the other hand, the Fisher test result demonstrated that the model is statistically significant, because F(6.83) = 10.8 > F_kr_. The value of coefficient p was substantially lower than 0.05. The estimation error for the entire model developed for the medium vibration frequency band amounted to SE_e_ = 1.832. When considering the results presented in Table 3, it is possible to see only a single factor that substantially affects the vibration generated in the frequency range of 300–1800 Hz, i.e., the inner ring race’s waviness deviation (RONt_PW:(16–50)_). This is demonstrated by the conducted student *t*-tests (see Table 3) and the high determination coefficient, which amounted to β = 0.7029.

When studying the data presented in the chart of Figure 7, it is possible to see a concentration of results in two areas of axis x. One of them concerns waviness deviations in area ${\mathrm{RONt}}_{\mathrm{PW}:\left(16-50\right)}\epsilon \langle 0.03;0.22\rangle$., while the other—in area ${\mathrm{RONt}}_{\mathrm{PW}:\left(16-50\right)}\epsilon \langle 0.37;0.79\rangle$. This is related to the fact that the testing featured a group of 90 bearings of the same type but from different manufacturers. Due to the above, the chart includes visible disproportion between particular batches. The first concentration features the results of bearings from two manufacturers. The other concentration features bearings from the third manufacturer, which points to a lower quality of the outer raceway’s surface. This can derive from an incorrectly performed finishing process (excess vibration of the grinding wheel, incorrect processing parameters). However, when analyzing the trend line in the chart of Figure 7 (red line), it is possible to see an upward trend in vibration along with the increase in the inner raceway’s waviness deviation. Similar results were obtained for the low vibration frequency range LB (Figure 5 and Figure 6).

When considering the mathematical model calculated for the high frequency band (HB), it can be stated that the highest determination coefficient among all tested models was obtained, i.e., R^2^ = 0.69. This means that 69% of the dependent variables were explained in this model. The good matching of the mathematical model and its statistical significance is also confirmed by the Fisher test, where F(6.83) = 30.66 > F_kr_. As with the model obtained for the medium vibration frequency band, the coefficient p was lower than 0.05. The estimation error for the entire model amounted to SE_e_ = 0.396. When analyzing the student *t*-test results and the coefficient p presented in Table 4, it is possible to see a statistical correlation of 4 input factors (RONt_PZ:(2–15)_, RONt_PW:(16–50)_, ΔR, f_t_) on the generated vibration recorded in the frequency range 1800 Hz–10,000 Hz. Due to the above, they will be analyzed in detail in the charts.

When analyzing the results presented in Figure 8, it is possible to see a small diversification between the measured outer ring’s roundness deviations. Most results do not exceed 2 µm, which indicates a stable manufacturing process maintained in all of the tested rolling bearing groups. There are only two exceptions where RONt_PZ:(2–15)_ = 2.04 µm and RONt_PZ:(2–15)_ = 5.86 µm were obtained. This may result from the analyzed ring’s factory defects. When analyzing all test results, it is possible to see discrepancies in the vibration characteristics despite the small variation of the inner ring’s roundness deviations. This indicates a dominant impact of other factors that can affect the generated vibration. The trend line presented in the chart points to a slight upward trend in the vibration recorded in the high frequency range due to the increase in the tested deviation.

When considering the results presented in Figure 9, it is possible to see a clear increase in the vibration generated in the high frequency band due to the increase in the inner ring’s waviness deviation (see the trend’s red line). Similarly, to Figure 7, it is possible to see a concentration of results in two intervals, which also results from the analysis of rolling bearings from different manufacturers. However, in the case of the high vibration frequency band, the trend line is more vertical, which points to a clearer impact of the inner ring’s waviness deviation on the generated vibration.

When testing the impact of radial clearance on the rolling bearings’ operation, it can be clearly stated that the increase in radial clearance increases the vibration generated in the frequency band of 1800–10,000 Hz. It is necessary to add that the testing featured bearings of the same type (6304ZZ) with a standard radial clearance. According to the ISO 5753-1:2009 standard [30], this type of bearings should have a radial clearance in the range of 0.005 mm–0.020 mm. It is possible to see that some bearings have a greater radial clearance, which allows for classifying them to the bearing group with the C3 clearance, i.e., bearings with greater radial clearance. It is necessary to note that the bearings with a greater radial clearance can operate at higher revolving speeds and have a lower friction moment. However, this has an adverse effect on the generated vibration, because excessive radial clearance allows the rolling elements to generate additional vibration during motion, which is visible in the chart of Figure 10. Due to the above, it is a significant factor visible in the high vibration frequency band.

When analyzing the results of testing the impact of the total curvature ratio on the vibration generated in the high frequency band presented in Figure 11, it is possible to see three measurement result groups, which is also related to the testing of bearings from three manufacturers. As can be seen from Dependency (14), the total curvature ratios derive from the inner and outer ring raceway’s radii and from the rolling balls’ diameter. In most cases, it is a parameter that is not standardized nor provided to users by the manufacturers. Hence the disproportions between the total curvature ratios obtained for different manufacturers. However, when analyzing the trend line, it can be stated that the increase in the total curvature ratio causes a moderate increase in the vibration generated in the frequency range of 1800–10,000 Hz. This is confirmed by the test presented in paper [26].

The test results in Table 5 were used to present the mathematical models specifying the impact of significant factors on the generated vibration. Only statistically significant factors were taken into consideration in these models.

When analyzing the mathematical models presented in Table 5, it can be stated that the most statistically significant predictors exists for the model developed for the high vibration frequency band. Due to the above, in terms of analyzing the impact of geometry on the quality of the rolling bearing race’s surface layer, this range should especially subjected to analysis. This was also confirmed by the statistical significance testing of the developed models, where the determination coefficient of R^2^ = 0.69 and estimation error of SE_e_ = 0.396 were obtained for HB. This points to the mathematical model’s good quality.

The innovation of mathematical models presented in Table 5 is the possibility of predicting the values of vibrations generated in specific frequencies based on the measured deviations and bearing parameters. Moreover, the developed models allow the producers of rolling bearings to indicate factors that may significantly affect the vibrations generated by the bearing type 6304zz. Then, it is possible to carry out a correction to the production process in order to obtain bearings of satisfactory quality.

## 4. Conclusions

The main purpose of the testing presented in the article was to evaluate the impact of the race’s roundness and waviness deviations, radial clearance and total curvature ratio on the vibration generated by 6304ZZ type rolling ball bearings made of AISI 52100 bearing steel. Mathematical models based on multi-dimensional regression equations were developed to conduct a quantitative evaluation of these parameters. The following conclusions were drawn based on the test results.

The literature analysis demonstrated the lack of papers that would include a quantitative evaluation of the impact of many factors on the vibration generated by rolling bearings.The total measurement results for bearings of the same type but offered by various manufacturers demonstrated clear disproportions between the measured parameters. This derives from the manufacturing quality and the use of treatment that in most cases were company secrets.The multi-dimensional regression equations can be effectively used for evaluating the impact of many factors on the vibration generated by rolling bearings.The increase in the race’s waviness and roundness deviations causes a clear increase in vibration.The increase in radial clearance causes an increase in vibration recorded in the high vibration frequency band, which may derive from the balls’ skidding on the race’s surface.The increase in the total curvature ratio causes increased vibration, which may derive from the reduction in friction between the balls’ surface and the race surface.For all of the analyzed models, the inner ring race’s waviness deviation turned out to be statistically significant. This is confirmed by the dominant impact of this deviation among all of the tested factors, and its impact should be minimized.The best matching of the model was obtained for the high vibration frequency band (1800 Hz–10,000 Hz). The model explained 69% of all independent variables. It also featured the lowest estimation error.The worst statistical results were obtained for the model developed for the medium frequency band. The inner raceway’s waviness deviation in the model had a dominant impact. It is therefore possible to state that other non-analyzed factors had a dominant impact in the vibration frequency range of 300 Hz–1800 Hz.Due to the fact that excessive values of bearing vibrations contribute to the propagation of the wear process of the bearings, it can be concluded that the waviness and roundness raceways deviations indirectly influence the bearing service life.

The direction of further research will be the analysis of other factors on the vibration generated by rolling bearings, e.g., contamination, type of lubricating agent used and resistance torque. Furthermore, it will include the testing of other methods aimed at evaluating the factors’ impact, e.g., application of artificial neuron networks.

## Figures and Tables

**Figure 1 materials-13-05440-f001:**
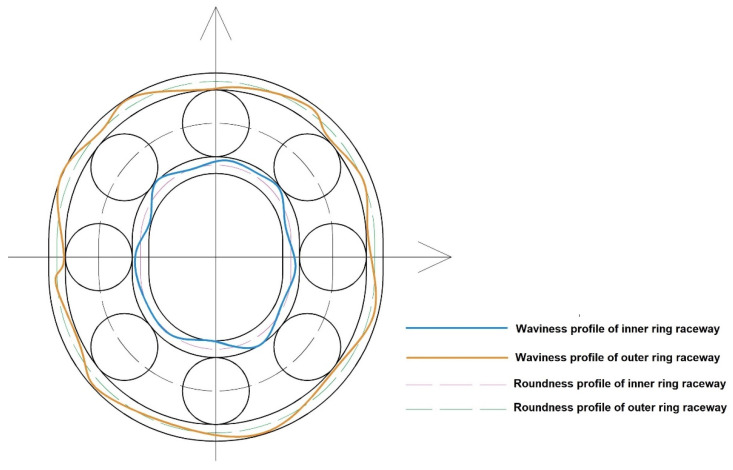
Example of roundness and waviness outlines on the rolling ball bearing raceways.

**Figure 2 materials-13-05440-f002:**
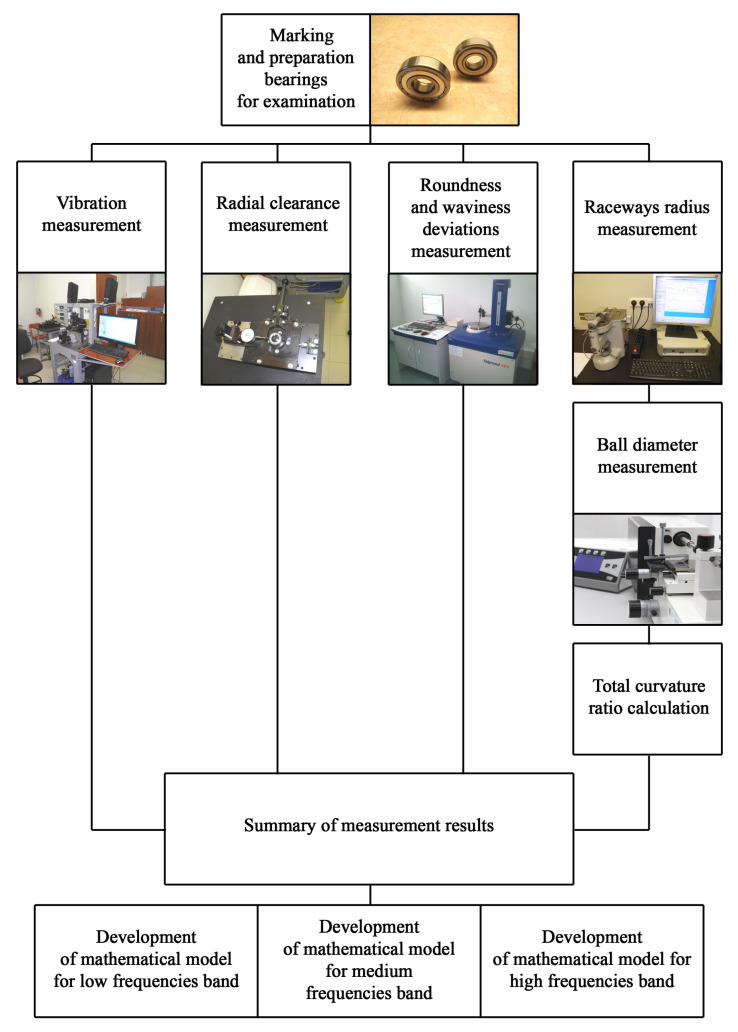
Particular stages of experimental testing.

**Figure 3 materials-13-05440-f003:**
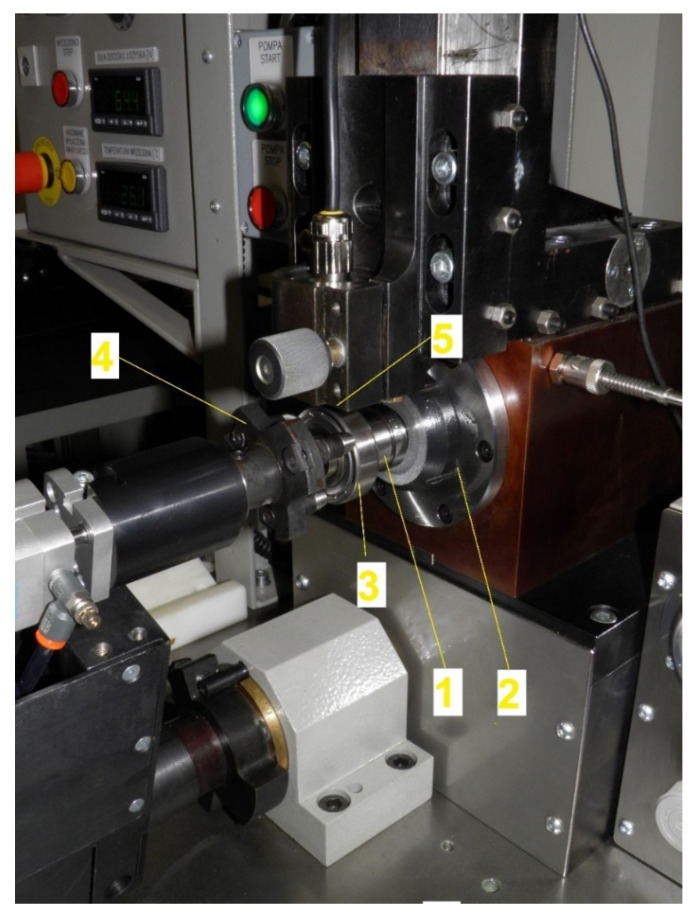
Test head of the Anderonmeter instrument.

**Figure 4 materials-13-05440-f004:**
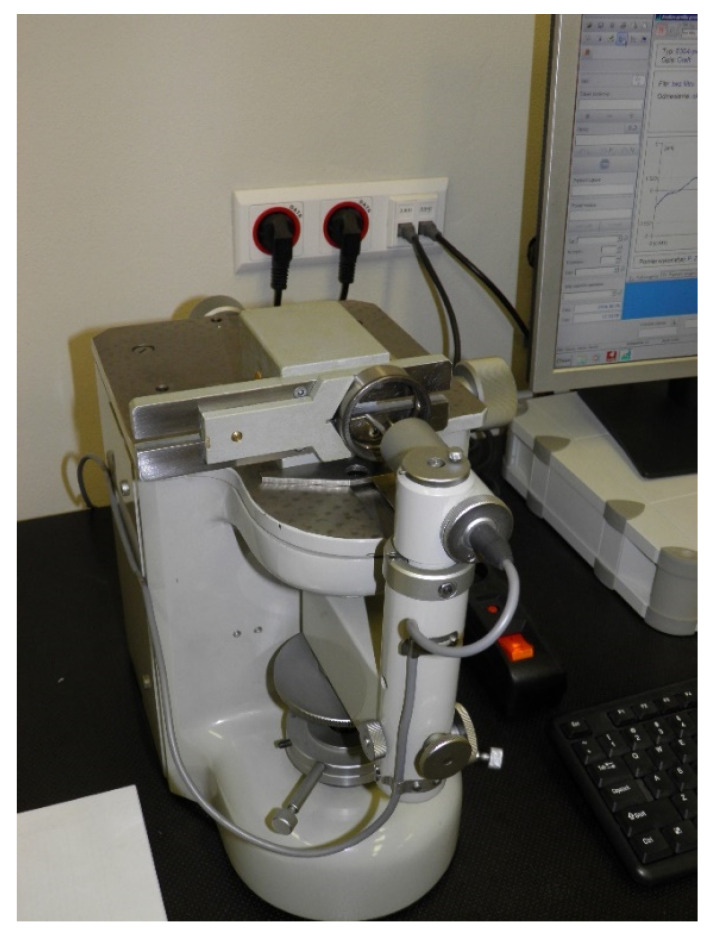
STPP instrument for measuring rolling bearing raceway geometries.

**Figure 5 materials-13-05440-f005:**
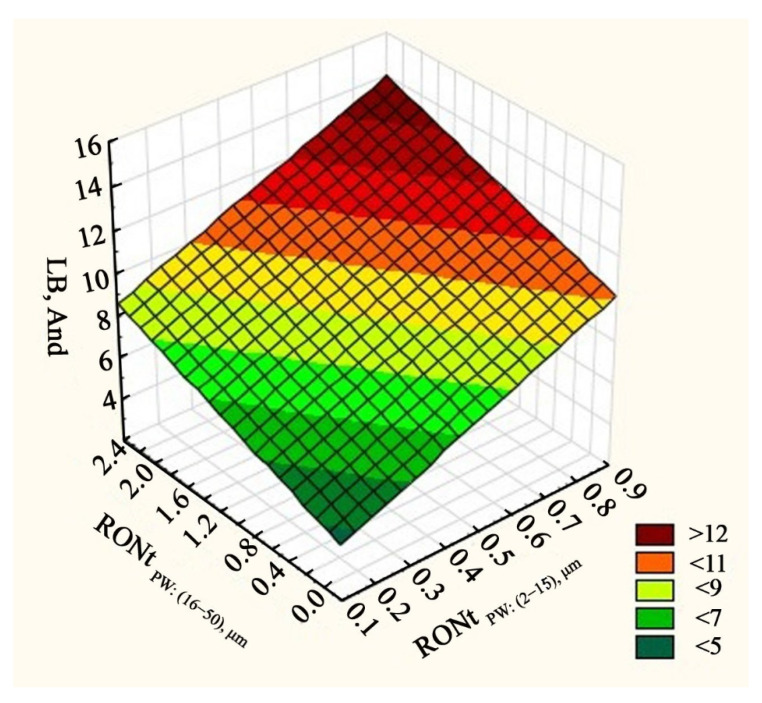
Impact of the inner ring raceway’s roundness and waviness deviation on the vibration (AND—Anderon Unit) generated in the LB frequency range.

**Figure 6 materials-13-05440-f006:**
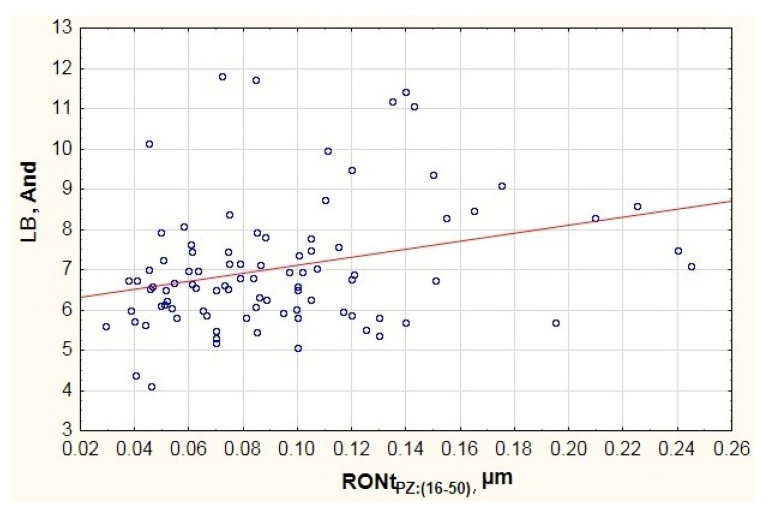
Impact of the inner ring raceway’s roundness and waviness deviation on the vibration (And—Anderon Unit) generated in the LB frequency range.

**Figure 7 materials-13-05440-f007:**
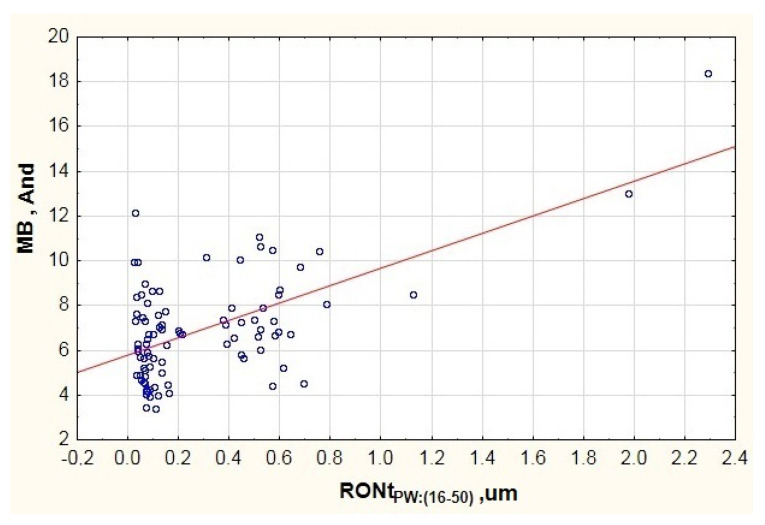
Impact of the inner ring race’s waviness deviation on the generated vibration (And—Anderon Unit) recorded in the medium frequency range MB.

**Figure 8 materials-13-05440-f008:**
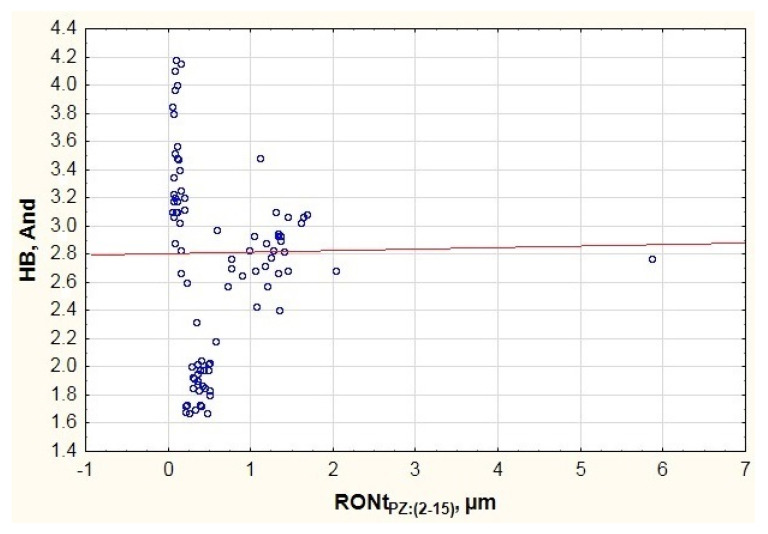
Impact of the outer ring raceway’s roundness deviation on the vibration (And—Anderon Unit) generated in the high frequency range HB.

**Figure 9 materials-13-05440-f009:**
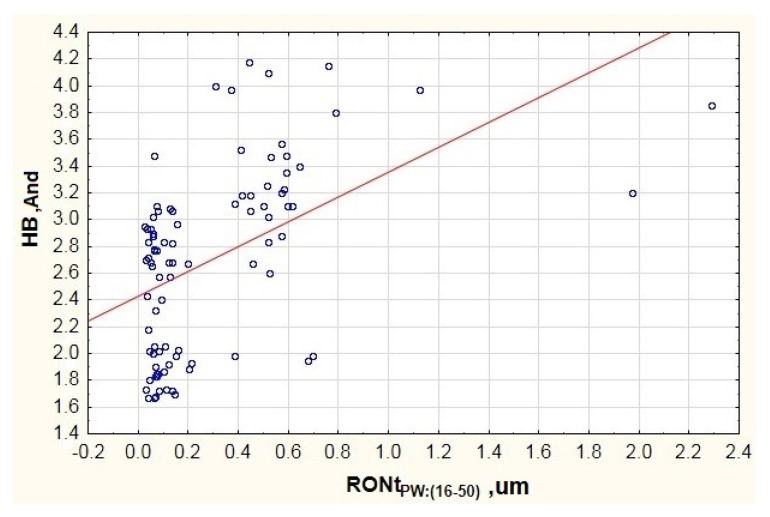
Impact of the inner ring raceway’s waviness deviation on the generated vibration (AND—Anderon Unit) recorded in the high frequency range HB.

**Figure 10 materials-13-05440-f010:**
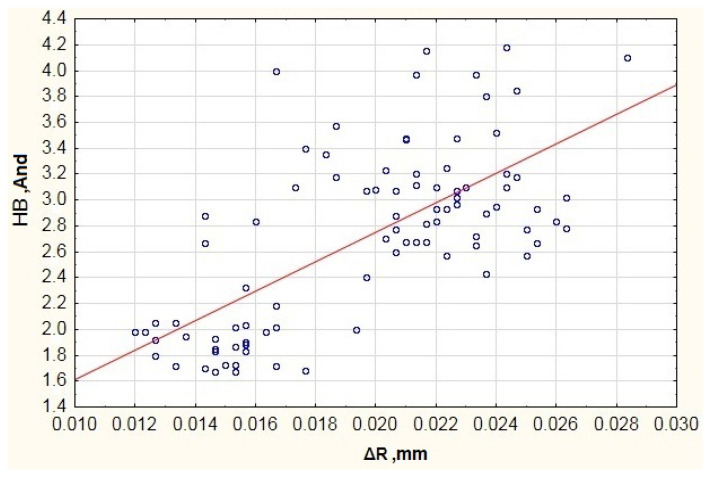
Impact of radial clearance on the generated vibration (And—Anderon Unit) recorded in the high frequency range HB.

**Figure 11 materials-13-05440-f011:**
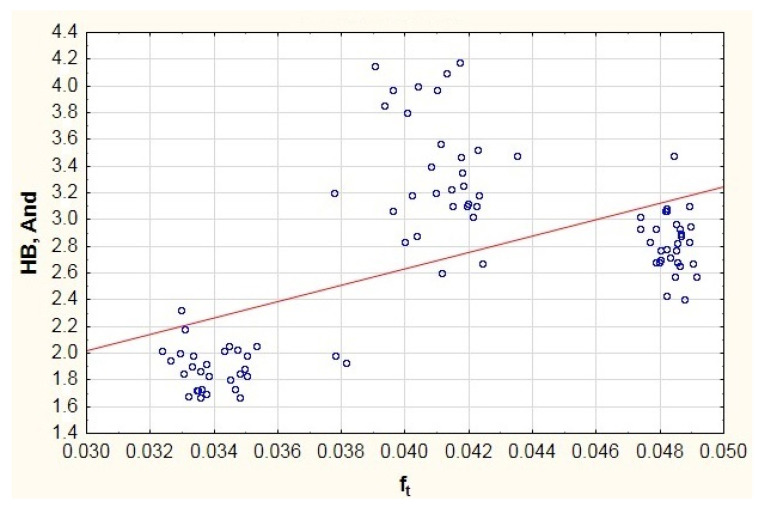
Impact of total curvature ratio on the generated vibration (And—Anderon Unit) recorded in the high frequency range HB.

**Table 1 materials-13-05440-t001:** Technical data of 6304zz bearings.

Manufacturer	A		B		C		
Dimensions	20 mm × 52 mm × 15 mm						
Ball number	7						
Mass	0.146 kg		0.145 kg		0.149 kg		
Dynamic load rating	16.9 kN		15.9 kN		16.8 kN		
Static load rating	7.9 kN		7.9 kN		7.8 kN		
Limiting speed	16,800 rpm		14,000 rpm		15,000 rpm		
Material	AISI 52100 bearing steel						
AISI 52100 chemical composition	Fe	C	Cr	Mn	Si	S	F
	95.5%	0.98%	1.3%	0.25%	0.15%	≤0.025%	≤0.025%

**Table 2 materials-13-05440-t002:** Regression summary for dependent variable: LB.

Parameter Name	b	SE_b_	t(83)	*p-*Value	β
Intercept	3.745	1.150	3.256	0.0016	
RONt_PW:(2–15)_	6.669	1.100	6.062	0.0000	0.5166
RONt_PZ:(2–15)_	0.153	0.217	0.703	0.4838	0.0740
RONt_PW:(16–50)_	1.504	0.412	3.650	0.0005	0.3555
RONt_PZ:(16–50)_	7.261	3.197	2.272	0.0257	0.2133
ΔR	−1.475	50.399	−0.029	0.9767	−0.0039
f_t_	−3.555	38.109	−0.093	0.9259	−0.0135

The red color indicate the factor is significant.

**Table 3 materials-13-05440-t003:** Regression summary for dependent variable: MB.

Parameter Name	b	Std.Err.	t(83)	*p*-Value	β
Intercept	−53.131	58.7542	−0.9043	0.3685	
RONt_PW:(2–15)_	−1.239	1.6584	−0.7470	0.4572	−0.0635
RONt_PZ:(2–15)_	−0.351	0.3484	−1.0076	0.3166	−0.1123
RONt_PW:(16–50)_	4.495	0.7450	6.0337	0.0000	0.7029
RONt_PZ:(16–50)_	2.490	4.8183	0.5168	0.6067	0.0484
ΔR	74.393	76.6679	0.9703	0.3347	0.1303
f_i_	−225.717	169.3096	−1.3332	0.1862	−0.2742

The red color indicate the factor is significant.

**Table 4 materials-13-05440-t004:** Regression summary for dependent variable: HB.

Parameter Name	b	Std.Err.	t(83)	*p*-Value	β
Intercept	−0.626	0.379	−1.65	0.103	
RONt_PW:(2–15)_	−0.075	0.363	−0.208	0.836	−0.013
RONt_PZ:(2–15)_	0.201	0.072	2.793	0.007	0.221
RONt_PW:(16–50)_	0.767	0.136	5.637	0.000	0.413
RONt_PZ:(16–50)_	−0.509	1.055	−0.482	0.631	−0.034
ΔR	58.533	16.637	3.518	0.001	0.353
f_i_	52.379	12.580	4.164	0.000	0.453

The red color indicate the factor is significant.

**Table 5 materials-13-05440-t005:** Equations for the regression of the impact of selected factors on the vibration generated in specific frequency ranges.

Dependent Variable	Regression Equation
LB	y = 3.745 + 6.669 RONt_PW:(2__–15)_ + 1.504 RONt_PW:(16–50)_ + 7.261 RONt_PZ:(16–50)_ + 1.197
MB	y = 4.495 RONt_PW:(16–50)_ + 1.832
HB	y = 0.201 RONt_PZ:(2–15)_ + 0.767 RONt_PW:(16–50)_ + 58.533 ΔR + 52.379 f_t_ + 0.396

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
