# Peer review of "Multi-Dimensional Mathematical Wear Models of Vibration Generated by Rolling Ball Bearings Made of AISI 52100 Bearing Steel"

_materials, 2020, doi:10.3390/ma13235440_

Round 1

Reviewer 1 Report

  1. Line 70: …operating bearings [4][5].-> …operating bearings [4,5].
  2. Line 186: …measurement signal analyses [26][27].-> …measurement signal analyses [26,27].
  3. Line 282: …?4?????:(16−50)++ ?5??.-> …?4??????:(16−50) + ?5??.
  4. Line 345, Table 1: 16800min-1, 14000min-1…-> Table 1: 16800rpm, 14000rpm…, or the other units.
  5. Line 496: please double check the sentence “?????:(16−50)?<0.03;0.22>, while the other – in area ??????:(16−50)?<0.373;0.786>”, or a detail description is necessary for the formulas.
  6. It is better to give a more comprehensible explanation about the formula (8), for example, the six factors. The reviewer suggests that the Figure 4 should be moved here to explain these factors, RONtpw:(2-15), RONtpz(2-15)…, and the meaning of 2-15 upr and 16-50 upr. The reviewer also thinks that the readability related to this is confused and failed, for example, lines 452-464.
  7. The meaning of every symbol should be explained when it is first introduced. Therefore, the meanings of “ro”, “ri”, and “Dw” on line 398, and “And” on Figures 5-8, and 11 must be identified.
  8. The measurements of the bearing’s vibration are important in this manuscript. Will the resonance frequency of the tested bearing avoid these three frequency bands: 50-300Hz (low), 300-1,800Hz (medium) and 1,800Hz – 10,000Hz (high)?
  9. All parameters must be fully explained in Table 2 for the sake of clarity.
  10. More emphasis should be stated on the innovation and its applications of the Regression equations shown on Table 5.

Author Response

Response to Reviewer 1 Comments

Dear Reviewer,

Thank you very much for reviewing our paper. Responding to Reviewer comments I prepared a revised version of the manuscript that include all remarks and suggestion.

Below I am sending the response to Reviewer comments:

Point 1: Line 70: …operating bearings [4][5].-> …operating bearings [4,5].

Response 1: The citations have been correct.

Point 2: Line 186: …measurement signal analyses [26][27].-> …measurement signal analyses [26,27].

Response 2: The citations have been corrected.

Point 3: Line 282: …?4?????:(16−50)++ ?5??.-> …?4??????:(16−50) + ?5??.

Response 3: The equation has been corrected.

Point 4: Line 345, Table 1: 16800min-1, 14000min-1…-> Table 1: 16800rpm, 14000rpm…, or the other units.

Response 4: The units in table 1 have been corrected.

Point 5: Line 496: please double check the sentence “?????:(16−50)?<0.03;0.22>, while the other – in area ??????:(16−50)?<0.373;0.786>”, or a detail description is necessary for the formulas.

Response 5: Thank you for your comment. Yes, it was a mistake, related to the rounding of value of measuring results. The values of the roundness and waviness deviation resulted from the calculation of the software and are presented in micrometers (µm). Therefore, reporting the results to three decimal places is pointless. It was corrected in the text.

Point 6: It is better to give a more comprehensible explanation about the formula (8), for example, the six factors. The reviewer suggests that the Figure 4 should be moved here to explain these factors, RONtpw:(2-15)RONtpz(2-15)…, and the meaning of 2-15 upr and 16-50 upr. The reviewer also thinks that the readability related to this is confused and failed, for example, lines 452-464.

Response 6:  Thank you very much for the good advice. The figure 4 and decryption of meaning of roundness and waviness deviation were moved to page 7. Moreover, detailed distribution of meaning of another factors (ΔR and ft) included in formula 8 were added in lines 337 - 340. The description presented in lines 452-464 has been edited.

Point 7: The meaning of every symbol should be explained when it is first introduced. Therefore, the meanings of “ro”, “ri”, and “Dw” on line 398, and “And” on Figures 5-8, and 11 must be identified.

Response 7:  The meaning of every symbol were explained in paragraph “nomenclature and abbreviations” at the beginning of this article, however additional information about meaning of symbols “ro”, “ri”, and “Dw” have been added in lines 453-456. Furthermore, the meaning of Anderon Unit symbol has been added in line 582 and in description of figures 5-8 and 11.

Point 8: The measurements of the bearing’s vibration are important in this manuscript. Will the resonance frequency of the tested bearing avoid these three frequency bands: 50-300Hz (low), 300-1,800Hz (medium) and 1,800Hz – 10,000Hz (high)?

Response 8:  Thank you very much for important comment.  It should be noted that bearing components (inner and outer ring, balls, cage) generate periodic frequencies called fundamental frequencies. Values of this specific frequencies in many cases is provided by bearing manufacturers. Therefore, in the study presented in paper, these frequencies have been input to Anderonmeter software and include in vibration analysis. 

Point 9: All parameters must be fully explained in Table 2 for the sake of clarity.

Response 9:  The measuring and statistical parameters were detailed described in subsection 2.1  Proposed method, however in order to clarity information in Table 2 additional description of statistical parameters have been added in lines 551-553. The measuring parameters presented in table 2 were described above table 2 (lines 545-548).

Point 10: More emphasis should be stated on the innovation and its applications of the Regression equations shown on Table 5.

Response 10:  Thank you very much for the good advice. The innovation of mathematical models presented in table 5 is the possibility of predicting the values of vibrations generated in specific frequencies based on the measured deviations and bearing parameters. Moreover, the developed models allow the producers of rolling bearings to indicate factors that may significantly affect the vibrations generated by the bearing type 6304zz. Then it is possible to carry out a correction to the production process in order to obtain bearings of satisfactory quality. This information has been added in the last paragraph of section 3 (lines 579-584).

I have a hope that my answers allowed explaining all questions posed by the Reviewer and revision version of my article would be accepted.

With best regards,

Paweł Zmarzły

Reviewer 2 Report

The submitted paper discusses about the multi-dimensional mathematical models used for evaluating the impact of selected factors on the vibration generated by rolling ball bearings. In my opinion, the paper has been fairly-organized and can be considered for publication. However, the following comments should be considered by authors before the publication of their manuscript in the journal.

  • I am wondering about the novelty of the paper. It should be clearly mentioned in the introduction section.
  • The introduction part is too long for a research paper. It should be shortened. Moreover, the objective of the paper should come at the last paragraph of the introduction.
  • In page 5, the authors implied that "the experiment planning methods should make the minimum number of samples amount to n=25=64". It should be considered that 2^5 equals 32 not 64.
  • Why did the authors choose the 6304ZZ type bearing for testing based on the market availability?
  • The chemical composition of the 6304ZZ type bearing should be presented in a Table.
  • The Anderonmeter measurement system used for vibration analysis should be explained in more detail.
  • It should be more clarified that how the use of the mandrel’s hydrodynamic bearing mounting eliminates the impact of own vibration.
  • In page 13, the authors implied that "For the entire mathematical model designated for the low vibration frequency band (LB) the determination coefficient of R2 = 0.46 was obtained”. The author should discuss about the R2 values which can show the development of model correctly.
  • The authors should discuss why the increase in the waviness deviation causes a moderate increase in the vibration recorded in the low frequency range.
  • There are some minor grammatical errors through the paper. Please also take a careful look and revise the quality of the English grammar and syntax where needed.

Author Response

Response to Reviewer 2 Comments

Dear Reviewer,

Thank you very much for reviewing our paper. Responding to Reviewer comments I prepared revised version of manuscript that include all remarks and suggestion.

Below I am sending the response to Reviewer comments:

Point 1: I am wondering about the novelty of the paper. It should be clearly mentioned in the introduction section.

Response 1: A novelty of the article is development multi-dimensional mathematical models used to quantitative analysis of the impact of selected factors on vibrations generated by rolling bearings made of AISI 52100 steel. In many scientific works only single factor is examined in full range of vibration frequencies.  Moreover, results obtained using mathematical models can provide guidance to manufacturers of rolling bearings, which type of bearing shape deviation should be minimized in order to produce the bearings of satisfactory quality. This information has been mentioned in two last paragraphs of introduction.

Point 2: The introduction part is too long for a research paper. It should be shortened. Moreover, the objective of the paper should come at the last paragraph of the introduction.

Response 2: Thank you very much for the good advice. The introduction has been rebuilt and shortened. The objective of the paper has been focused at the last paragraph of the introduction.

Point 3: In page 5, the authors implied that "the experiment planning methods should make the minimum number of samples amount to n=25=64". It should be considered that 2^5 equals 32 not 64

Response 3: Thank you very much for attention. Yes, it was mistake in text. In the experiment planning there are 6 different variables dependent: , , , ΔR, ft . Therefore, it should be n=26=64. This mistake has been corrected in text.

Point 4: Why did the authors choose the 6304ZZ type bearing for testing based on the market availability?

Response 4: Bearing type 6304ZZ is one of the most popular ball bearings applied in automotive and industrial machines. This type of the bearing is sealed with plates, thereby eliminating the risk of contaminating the lubricating.

Point 5: The chemical composition of the 6304ZZ type bearing should be presented in a Table.

Response 5:  The chemical composition of the 6304ZZ type bearing has been added in table 1.

Point 6: The Anderonmeter measurement system used for vibration analysis should be explained in more detail.

Response 6: The more detailed information related to Anderonmeter has been added in lines 362-367.

Point 7: It should be more clarified that how the use of the mandrel’s hydrodynamic bearing mounting eliminates the impact of own vibration.

Response 7:  Thank you for good comment. I agree that the description of the hydrodynamic bearing should be clarified. In the text is some mistake. In measuring system Anderonmeter the measured bearing (3) is embedded on the roller (1). The roller (1) with the measuring bearing (3) is embedded in the spindle equipment with hydrodynamic bearing. This bearing is lubricated with oil mist from exterior pomp. In the many precision measuring devices the hydrostatic or hydrodynamic bearing are used. This types of the bearing have a simple construction and does not have balls that can generate additional vibrations. The description of hydrodynamic bearing has been corrected in the paper (lines 354-358).

Point 8: In page 13, the authors implied that "For the entire mathematical model designated for the low vibration frequency band (LB) the determination coefficient of R2 = 0.46 was obtained”. The author should discuss about the R2 values which can show the development of model correctly.

Response 8:  The relatively small value of the determination coefficient R2 indicates that there may be other factors (not analyzed in the tests) affecting the vibration values recorded in this frequency band. It should be noted that that the value of vibrations generated in the low frequency range is influenced by difficult-to-measure factors, such as: cage unbalance, contamination of the lubricant, etc. This information has been added in paper (lines 440-445).

Point 9: The authors should discuss why the increase in the waviness deviation causes a moderate increase in the vibration recorded in the low frequency range.

Response 9:  Waviness deviation is periodic or non-periodic irregularities appearing on bearing raceway profile. The largest waviness deviation indicated highest wave’s amplitude on the raceway profile. Therefore, during working, bearing balls generate higher vibration value, which is also recorded in the low frequency range.  This information was described in lines 470-473.

Point 10: There are some minor grammatical errors through the paper. Please also take a careful look and revise the quality of the English grammar and syntax where needed

Response 10: The English grammar and syntax of all text in the paper have been checked and corrected

I have a hope that my answers allowed explaining all questions posed by Reviewer and revision version of my article would be accepted.

With best regards,

Paweł Zmarzły

Round 2

Reviewer 2 Report

The revisions are satisfactory. The paper can be accepted for publication in the current format.